# Omega-3 Fatty Acids Modify Drp1 Expression and Activate the PINK1-Dependent Mitophagy Pathway in the Kidney and Heart of Adenine-Induced Uremic Rats

**DOI:** 10.3390/biomedicines12092107

**Published:** 2024-09-15

**Authors:** Dong Ho Choi, Su Mi Lee, Bin Na Park, Mi Hwa Lee, Dong Eun Yang, Young Ki Son, Seong Eun Kim, Won Suk An

**Affiliations:** 1Department of Internal Medicine, Good Moon Hwa Hospital, Busan 48735, Republic of Korea; 2Department of Internal Medicine, Dong-A University, Busan 49201, Republic of Korea; sumilee@dau.ac.kr (S.M.L.); binna1639@gmail.com (B.N.P.); eastin@dau.ac.kr (D.E.Y.); kidney@dau.ac.kr (Y.K.S.); sekim@dau.ac.kr (S.E.K.); 3Department of Anatomy and Cell Biology, Dong-A University, Busan 49201, Republic of Korea; hero15p@nate.com; 4Medical Science Research Center, Dong-A University, Busan 49201, Republic of Korea

**Keywords:** biogenesis, chronic kidney disease, dynamics, fatty acid, mitochondria, omega-3

## Abstract

Mitochondrial homeostasis is controlled by biogenesis, dynamics, and mitophagy. Mitochondrial dysfunction plays a central role in cardiovascular and renal disease and omega-3 fatty acids (FAs) are beneficial for cardiovascular disease. We investigated whether omega-3 fatty acids (FAs) regulate mitochondrial biogenesis, dynamics, and mitophagy in the kidney and heart of adenine-induced uremic rats. Eighteen male Sprague Dawley rats were divided into normal control, adenine control, and adenine with omega-3 FA groups. Using Western blot analysis, the kidney and heart expression of mitochondrial homeostasis-related molecules, including peroxisome proliferator-activated receptor gamma coactivator-1 alpha (PGC-1α), dynamin-related protein 1 (Drp1), and phosphatase and tensin homolog-induced putative kinase 1 (PINK1) were investigated. Compared to normal, serum creatinine and heart weight/body weight in adenine control were increased and slightly improved in the omega-3 FA group. Compared to the normal controls, the expression of PGC-1α and PINK1 in the kidney and heart of the adenine group was downregulated, which was reversed after omega-3 FA supplementation. Drp1 was upregulated in the kidney but downregulated in the heart in the adenine group. Drp1 expression in the heart recovered in the omega-3 FA group. Mitochondrial DNA (mtDNA) was decreased in the kidney and heart of the adenine control group but the mtDNA of the heart was recovered in the omega-3 FA group. Drp1, which is related to mitochondrial fission, may function oppositely in the uremic kidney and heart. Omega-3 FAs may be beneficial for mitochondrial homeostasis by activating mitochondrial biogenesis and PINK1-dependent mitophagy in the kidney and heart of uremic rats.

## 1. Introduction

Chronic kidney disease (CKD) is a progressive disease affecting 14% of the general population in the United States [1]. It is associated with an increased prevalence of cardiovascular disease, which is the main cause of morbidity and mortality in CKD patients [2,3]. Kidney dysfunction occurs frequently in patients with heart failure (HF) and is a risk factor for adverse cardiovascular outcomes. The relationship between kidney disease and HF is reflected through cardiorenal syndrome [4,5]. In accordance with the 2022 US Renal Data System Annual Data report, cardiovascular diseases are 2–3 times as common among individuals with CKD as among those without. Moreover, the prevalence of HF in patients with CKD aged 66 years and older is close to 24% compared to 6% in patients without CKD [1]. In accordance with the 2022 Korea Health Statistics, the prevalence of CKD aged over 65 years is close to 18% compared to 7% in total adult patients with CKD aged over 18 years [6]. Moreover, the prevalence of HF with CKD was 15.8% in the total Korean population in 2020, compared to 2.58% in patients without CKD [7]. Therefore, the kidney and heart exhibit a close bidirectional association.

Mitochondria are energy-producing organelles that maintain various cellular functions. Mitochondrial homeostasis requires a balance between biogenesis, dynamics (fission and fusion), and mitophagy. Its dysfunction can play a central role in the development and progression of cardiovascular and kidney disease [8,9]. Peroxisome proliferator-activated receptor gamma coactivator-1 alpha (PGC-1α), a regulator of mitochondrial biogenesis, is abundant in the kidney and heart due to high energy demand [10,11]. PGC-1α expression in CKD experimental mice was downregulated and its expression showed a positive effect against oxidative stress and kidney fibrosis [12]. In addition, the downregulation of PGC-1α facilitates the pathogenesis and progression of HF [11]. Mitochondrial fission, involving dynamin-related protein 1 (Drp1), segregates damaged mitochondria. Increased Drp1 expression has also been observed in kidney diseases [8,13,14]. In contrast, cardiomyocyte-specific conditional Drp1 knockout mice developed HF and died after 8–13 weeks [15]. Mitophagy selectively degrades excess or defective mitochondria. In experimental models, such as unilateral ureteral obstruction (UUO) mice, the deletion of phosphatase and tensin homolog (PTEN)-induced putative kinase 1 (PINK1) exacerbated kidney injury, indicating that mitophagy plays a protective role in CKD [16,17].

Omega-3 fatty acids (FAs) have anti-inflammatory and renoprotective effects against kidney diseases by mitigating oxidative stress and inflammation. Moreover, a high level of omega-3 FAs was associated with a slower annual decline in renal function [18,19,20]. They reduce cardiovascular risk through various mechanisms, including anti-inflammatory action, the lowering of triglyceride-rich lipoproteins, and antithrombotic effects, by suppressing the expression of pro-inflammatory cytokines and the infiltration of inflammatory cells [21].

To the best of our knowledge, there are no reports on the expression of markers related to mitochondrial biogenesis, dynamics, and mitophagy in both the kidney and heart of experimental CKD models. We investigated whether omega-3 FAs regulated mitochondrial biogenesis, dynamics, and mitophagy in the kidney and heart of adenine-induced uremic rats.

## 2. Materials and Methods

### 2.1. Animals and Experimental Design

Eighteen male Sprague Dawley rats were obtained from Central Lab. Animals Inc. (Seoul, Republic of Korea), and randomly divided into three groups. Group 1 (n = 6) consisted of normal control rats, which were administered diets containing 2.5% protein and saline (1 mL/kg per day by gastric gavage) for 7 weeks. Group 2 (n = 6) consisted of adenine control rats. After feeding a 0.75% adenine and 2.5% protein diet for the first 3 weeks to induce uremia, they were administered a 2.5% protein diet and saline (1 mL per kg per day by gastric gavage) for the next 4 weeks. Group 3 (n = 6) consisted of adenine with omega-3 FA rats. After feeding a 0.75% adenine and 2.5% protein diet for the first 3 weeks to induce uremia, they were administered a 2.5% protein diet with omega-3 FA (Omacor^®^, 300 mg per kg per day by gastric gavage) for the next 4 weeks. Omacor^®^, marine-originated omega-3 FA, is made up of 460 mg eicosapentaenoic acid and 380 mg of docosahexaenoic acid in 1000 mg of Omacor^®^ [22]. All procedures involving animals were performed with the approval of the Institutional Animal Care Committee of Dong-A University (DIACUC-16-9) and conducted in accordance with the Guide for the Care and Use of Laboratory Animals published by the US National Institutes of Health.

### 2.2. Western Blotting and Immunoprecipitation

Western blotting was performed as previously described, with slight modifications [10]. The antibodies against PGC-1α, SIRT 1/3, PINK1, BNIP3, Mfn1/2, and NIX were purchased from Santa Cruz Biotechnology (Santa Cruz, CA, USA). The antibodies against Nrf2 were obtained from Abcam (Cambridge, MA, USA). Anti-Drp1 and OPA1 antibodies were purchased from BD Bioscience (Franklin Lakes, NJ, USA). Antibodies against β-actin were purchased from Sigma-Aldrich (St. Louis, MO, USA).

Immunostaining with antibodies was performed using SuperSignal West Pico chemiluminescence substrate (Thermo Scientific, Hudson, NH, USA) and detected using AMERSHAM ImageQuant 800 (GE Healthcare Bio-Sciences AB, Uppsala, Sweden). Quantification and normalization to the β-actin control were conducted using ImageJ software (version 1.48q, National Institutes of Health, Bethesda, MD, USA).

### 2.3. Measurement of mtDNA Content

Quantitative real-time polymerase chain reaction (qRT-PCR) was used to determine the relative mtDNA content. Total RNA was extracted from tissues using TRIzol^®^ reagent. Then, 1 μg of total RNA was converted into single-stranded complementary DNA (cDNA) using a moloney murine leukemia virus cDNA synthesis kit (Enzynomics, Daejeon, Republic of Korea). Primers were designed from the respective gene sequences using Primer3 and mFold software version 0.4.0. For qRT-PCR analysis, cDNA was subjected to PCR amplification using gene-specific primers: rat mtDNA, 5′-GGT TCT TAC TTC AGG GCC ATC A-3′ (sense), 5′-TGA TTA GAC CCG TTA CCA TCG A-3′ (antisense); rat GAPDH, 5′-CAA GAA GGT GAA GCA GG-3′ (sense), 5′-GGT GGA AGA ATG GGA GTT GC-3′ (antisense). Real-time PCR was performed using SYBR Green PCR Master Mix (Applied Biosystems, Foster City, CA, USA) with an ABI 7500 instrument (Applied Biosystems, Waltham, MA, USA). The Master Mix contains SYBR Green dye, Dual-Lock Taq DNA Polymerase, dNTPs with dUTP/dTTP blend, heat-labile UDG, ROX passive reference dye, and optimized buffer components. The PCR condition for standard cycling was set according to the SYBR Green Master Mix protocol and guidance. 

### 2.4. Statistics

Statistical significance and all analyses among experimental groups were evaluated using SPSS software (version 18.0; IMB Corp., Armonk, NY, USA). All quantitative data are presented as mean ± standard deviation and were analyzed using the Mann–Whitney U test or Kruskal–Wallis test. A *p* value of less than 0.05 was considered statistically significant.

## 3. Results

### 3.1. Laboratory Data

The results obtained at 7 weeks are summarized in Table 1. Blood urea nitrogen (BUN), creatinine, and phosphorus levels were higher in the adenine and omega-3 FA groups than in the control group at 7 weeks after treatment. However, the BUN, creatinine, and phosphorus levels did not differ between the adenine and omega-3 FA groups but were slightly improved. The kidney weight-to-body weight ratio and heart weight-to-body weight ratio of the adenine and omega-3 FA groups were higher than those of the control group. Compared with the adenine group, the kidney weight-to-body weight ratio and heart weight-to-body weight ratio were not significantly different in the omega-3 FA group but were slightly improved.

### 3.2. Changes in Factors Related to Mitochondrial Biogenesis, Dynamics, and Mitophagy

#### 3.2.1. Mitochondrial Biogenesis-Related Molecules

The Western blotting of mitochondrial biogenesis-related molecules, including PGC-1α, sirtuin (SIRT) 1/3, and nuclear factor erythroid-2 related factor 2 (Nrf2), is shown in Figure 1. The expression of PGC-1α in the kidney and heart of the adenine group was significantly decreased compared to controls. In the adenine with omega-3 FA group, the expression of PGC-1α was increased, but it did not reach the level of the control group. Compared with controls, the adenine group showed a decreased expression of SIRT1/3 in the kidney (Figure 1A) and heart. However, only the adenine with omega-3 FA group showed a significant increase in SIRT1 expression in the heart (Figure 1B). The adenine group showed a decreased Nrf2 expression compared to the normal control group. In the adenine with omega-3 FA group, Nrf2 expression was higher than that in the adenine group; however, the difference was not statistically significant.

#### 3.2.2. Mitochondrial Fusion- and Fission-Related Molecules

The Western blotting for mitochondrial fusion- and fission-related molecules, including Drp1, optic atrophy 1 (OPA1), and mitofusin-1/2 (Mfn1/2), is shown in Figure 2. The expression of Drp1 and Mfn1/2 in the kidney of the adenine group was higher than that in the control group. In the kidney of the adenine with omega-3 FA group, it was lower than that in the adenine group. The expression of OPA1 in the kidney of the adenine group was lower than that in the control group. However, following supplementation with omega-3 FA, OPA1 expression did not significantly increase, and the difference between the groups was not statistically significant (Figure 2A).

The expression of Drp1, OPA1, and Mfn1/2 decreased in the heart of mice in the adenine group, whereas Drp1 and Mfn2 expression increased only after omega-3 FA supplementation (Figure 2B).

#### 3.2.3. Mitochondrial Mitophagy-Related Molecules

The Western blotting for mitochondrial mitophagy-related molecules, including PINK1, bcl-2/adenovirus E1B 19 kDa interacting protein-3 (BNIP3), and Nip3-like protein X (NIX), is shown in Figure 3. Compared with controls, the adenine group showed decreased PINK1 expression in the kidney. After omega-3 supplementation, the expression of PINK1 increased compared to that of the adenine group. In contrast, the expression of BNIP3 and NIX increased in the kidney of the adenine group. These values decreased after omega-3 FA supplementation (Figure 3A).

Compared to controls, the adenine group exhibited lower levels of PINK1, BNIP3, and NIX in the heart, which were significantly increased by omega-3 FA supplementation (Figure 3B).

#### 3.2.4. Content of Mitochondrial DNA (mtDNA)

mtDNA levels in the kidney and heart were lower in the adenine group than in the control group. The mtDNA level significantly increased only in the heart of the adenine with omega-3 FA group (Figure 4).

## 4. Discussion

Drp1 is a small guanosine triphosphatase (GTPase) that medicates mitochondrial fission. Drp1 was inversely expressed in the kidney and heart in this study using an adenine-induced CKD model. Furthermore, Drp1 expression increased in the heart and decreased in the kidney after omega-3 FA supplementation. Several studies have shown that excessive fission is mediated by Drp1 overexpression in kidney diseases [8,13,14]. Similarly, in this study, the expression of Drp1 was increased in the kidney of the adenine group. Therefore, the effect of omega-3 FA in reducing Drp1 expression in the kidney may be beneficial in CKD models, although this effect was not prominent in this study. Likewise, previous studies using Drp1 knockout mice have shown that Drp1 downregulation induced mitochondrial dysfunction, apoptosis, left ventricular dysfunction, and heart failure [15,23]. In this study, Drp1 in the heart of adenine-induced CKD mice was downregulated, and mitigated after omega-3 FA supplementation. 

Mitochondrial fusion is mediated by the outer mitochondrial membrane (OMM) protein Mfn1/2 and the inner mitochondrial membrane (IMM) protein, OPA1 [24]. Mfn1 exhibits higher GTPase activity than Mfn2, and OPA1 cannot induce mitochondrial fusion under Mfn1 deficiency. In contrast, Mfn2 is related to mitochondrial mitophagy and bridges PINK1 and Parkin [25]. In a recent study using Mfn2 knockout UUO mice, its deficiency aggravates kidney fibrosis [16]. In addition, under metabolic stress conditions, such as ATP depletion, Mfn2 deficiency aggravates kidney epithelial cell injury [26]. Therefore, the increased expression of Mfn2 in the kidney may be related to a compensatory mechanism to decrease the aggravation of kidney fibrosis and kidney injury in the adenine group. In this study, Mfn2 levels were upregulated in the kidney of the adenine group compared with those in the normal control group. After omega-3 FA administration, the level recovered compared to that in the adenine group. Mfn2 expression is downregulated in cardiac hypertrophy; however, Mfn upregulation via viral mRNA modulates myocyte hypertrophy [27]. Similar to Drp1, Mfn2 levels in the heart were reversed compared to those in the kidneys. OPA1 is a membrane-bound GTPase involved in IMM fusion [28]. A HF rat model showed decreased OPA1 expression, resulting in mitochondrial apoptosis [29]. OPA1 expression was downregulated, suggesting decreased mitochondrial fusion in the kidney of a diabetic mouse model [30]. In this study, OPA1 expression was not improved by omega-3 FA supplementation. Therefore, omega-3 FA may be more closely related to mitochondrial fission and OMM fusion than to IMM fusion in the kidneys and heart of the CKD model.

Mitophagy is a type of selective autophagy that eliminates damaged mitochondria and is modulated by several regulators including PINK1, Parkin, BNIP3, and NIX [31,32]. Damaged mitochondria induce the production of mitochondrial reactive oxygen species (ROS) which are removed by mitophagy [33]. Mitophagy is regulated by PINK1- and Parkin-dependent pathways, as well as by the BNIP3 and NIX pathways [34,35]. In this study, the expression of PINK1 in the kidney and heart decreased in adenine-induced uremic rats and recovered after omega-3 FA supplementation. Notably, BNIP3 and NIX were inversely expressed in the kidney and heart of the adenine group, although they recovered in the adenine with omega-3 FA group. First, the suppression of the PINK1 pathway may be harmful to the kidney and heart of the CKD model, and omega-3 FA can recover PINK1-dependent mitophagy. Based on this study, PINK1-dependent mitophagy is beneficial for eliminating damaged mitochondria in the kidney and heart. Second, BNIP3/NIX-dependent mitophagy activation shows that the kidney is severely injured whereas mitochondrial wastes are being removed well in the heart. Several studies have shown that the downregulation of PINK1 and BNIP3/NIX in a genetic intervention model induces pathological cardiac hypertrophy and cardiomegaly [36,37]. Biopsied heart samples from patients with HF and experimental models of HF showed decreased mitophagy and reduced PINK1 phosphorylation, which impaired mitochondrial function in a mouse model of HF [38,39]. In UUO mice, the deletion of PINK1 exacerbated kidney injury through mitochondrial ROS production, mitochondrial damage, and kidney fibrosis, suggesting that mitophagy plays a protective role in CKD [16,17].

Hypoxic stress stimulates BNIP3 and NIX expression via a mechanism involving hypoxia-inducible factor (HIF)-1α that directly induces the transcription of BNIP3 and NIX [40,41]. A recent study using human tubular cells and renal tubular-specific HIF-1α knockout mice showed that HIF-1α knockout inhibited mitophagy and increased ROS production and kidney damage [42]. Similarly, in the present study, BNIP3 and NIX were upregulated in the kidney of the adenine group and recovered after omega-3 FA supplementation. Recent studies have shown that BNIP3 and NIX induce the selective removal of mitochondria in cardiac myocytes [43,44]. Regarding waste removal, omega-3 FA supplementation induces BNIP3 and NIX expression and triggers mitophagy. Further studies are necessary to understand why PINK1-dependent mitophagy and BNIP3/NIX-dependent mitophagy differ in the kidney of CKD models. In terms of the double-sided impact, omega-3 FA supplementation may be effective in recovering mitophagy-related molecules in the kidney and heart of adenine-induced uremic rats.

PGC-1α is well known as a master regulator of mitochondrial biogenesis, mitochondrial dynamics, and mitophagy. Low PGC-1α expression in acute kidney injury (AKI) and CKD has been observed, while PGC-1α overexpression was related to a protective effect in vivo and in vitro AKI models (nephrotoxic, ischemic-reperfusion, and septic) [10,45,46,47]. Similar to previous studies, we found a significantly decreased expression of PGC-1α in the kidney of the adenine-induced uremic model. After omega-3 FA supplementation, the expression of PGC-1α increased but did not recover to the same level as the normal control group. There are studies suggesting that omega-3 FA increases the expression of PGC-1α, AMP-activated protein kinase, and SIRT in vivo (myoblast) and in vitro models (skeletal muscle cell) [48,49]. Our previous study showed that omega-3 FA may improve mitochondrial biogenesis by upregulating Nrf1 and Nrf2. The probable mechanism was related to increased PGC-1α expression and the deacetylation of PGC-1α, which was triggered by increased SIRT1/3 production [22]. In addition, PGC-1α deficiency leads to the development and progression of HF [11]. Several studies related to the cardiac-specific PGC-1α knockout model showed decreased cardiac contractile function and increased left ventricle fibrosis, which is a sign of HF [50,51]. In this study, the expression of PGC-1α and SIRT1 was decreased in the heart of the adenine group and recovered in the adenine with omega-3 FA group. Furthermore, the adenine group showed decreased mtDNA levels, which increased following omega-3 FA supplementation. PGC-1α upregulation and Nrf2 activation induced mtDNA replication and mitochondrial biogenesis [10,52]. A recent study showed that omega-3 FA has beneficial effects on cardiac function by regulating inflammation [53]. Therefore, omega-3 FA may improve mitochondrial biogenesis modulating PGC-1α and mtDNA.

There are several limitations to this study. First, we did not evaluate the reactive oxygen species (ROS) level and inflammatory markers related to the oxidative change at the mitochondrial levels in this study. Second, we did not perform a pathologic examination to assess the usefulness of omega-3 FA in this CKD model. Third, we did not investigate the glycolytic regulation and dysregulation related to mitochondrial dysfunction and mitochondrial homeostasis [54].

In conclusion, our results suggest that Drp1, which is involved in mitochondrial fission, may be an important target for management in the uremic kidney and heart. Omega-3 FAs may be beneficial for mitochondrial homeostasis by activating mitochondrial biogenesis and PINK1-dependent mitophagy in the kidney and heart of adenine-induced uremic rats. Further studies including mitochondrial function, ROS, and glycolytic regulation are necessary to demonstrate that omega-3 FAs may be beneficial for preventing the progression of renal function decline and heart failure. 

## Figures and Tables

**Figure 1 biomedicines-12-02107-f001:**
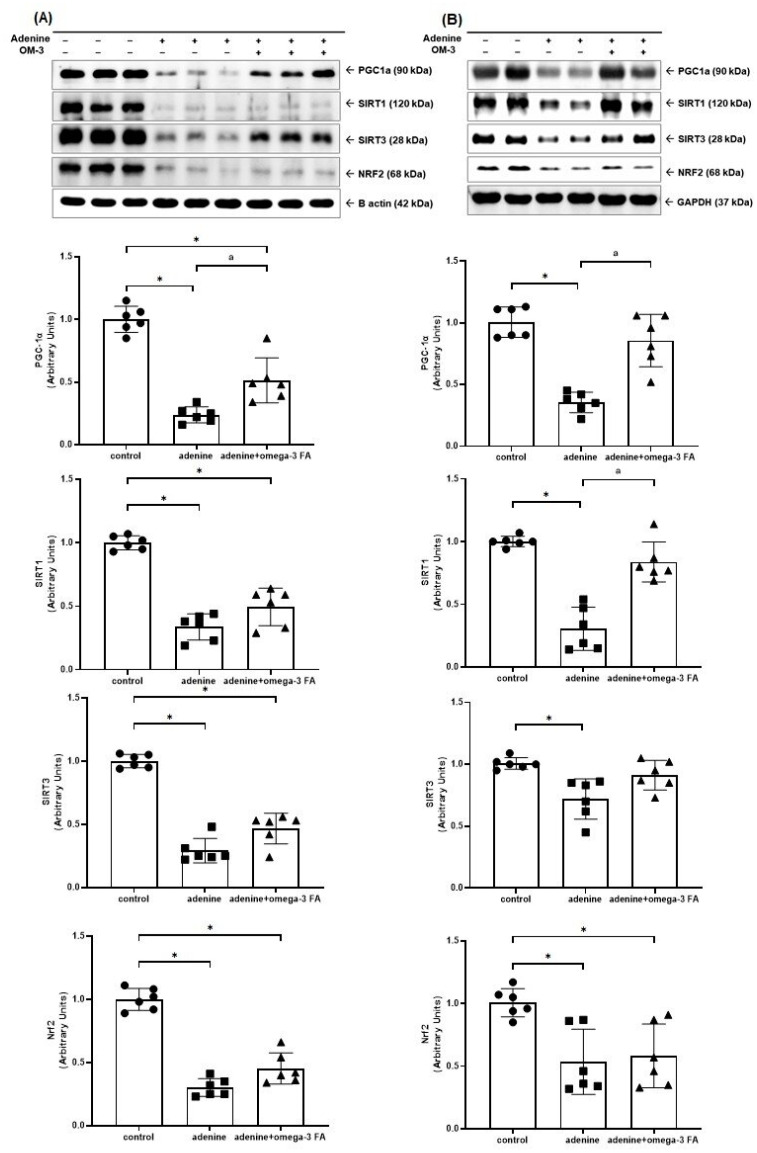
Changes in the expression of factors related to mitochondrial biogenesis including PGC-1α, SIRT1/3, and Nrf2 in the kidney (**A**) and heart (**B**). * *p* value < 0.05 (mean values are significantly different from the control group). ^a^
*p* value < 0.05 (mean values are significantly different from the adenine group).

**Figure 2 biomedicines-12-02107-f002:**
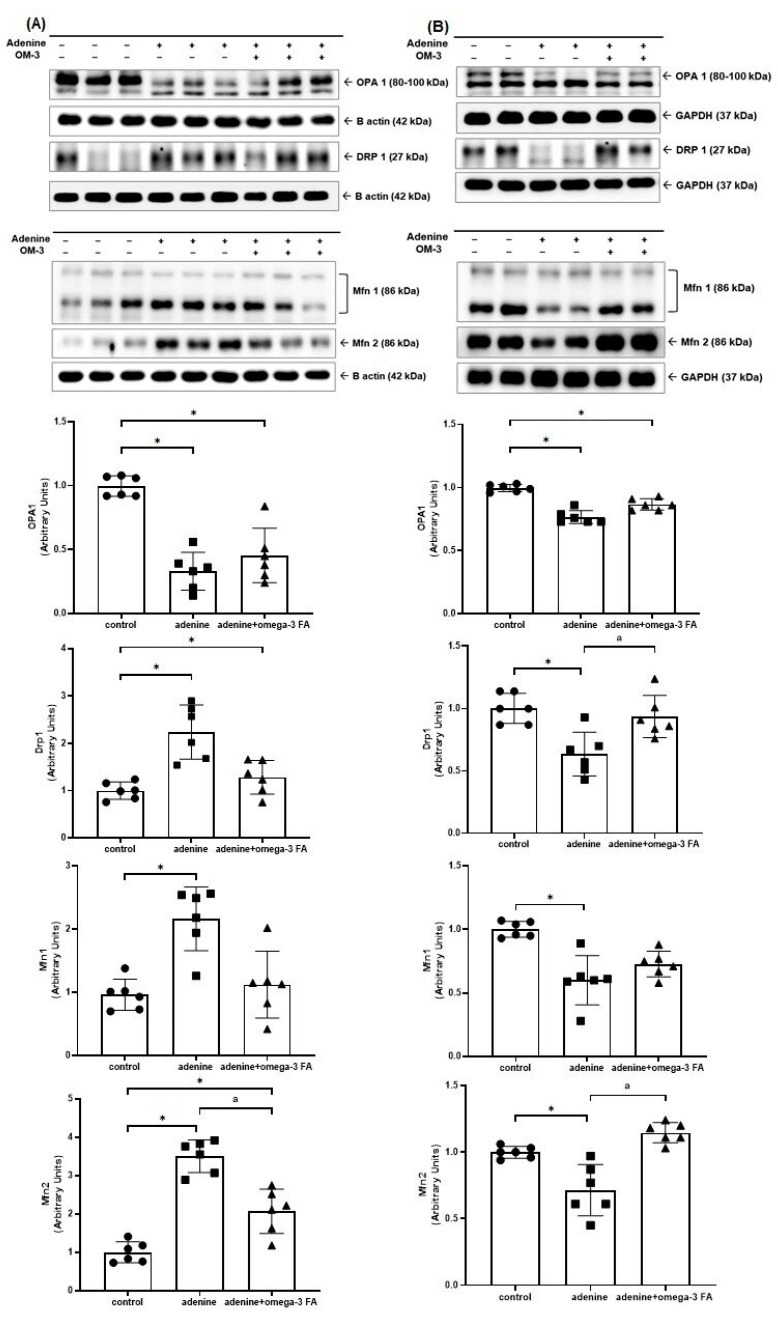
Changes in the expression of factors related to mitochondrial fusion and fission including OPA1, Drp1, and Mfn1/2 in the kidney (**A**) and heart (**B**). * *p* value < 0.05 (mean values are significantly different from normal control group). ^a^
*p* value < 0.05 (mean values are significantly different from the adenine group).

**Figure 3 biomedicines-12-02107-f003:**
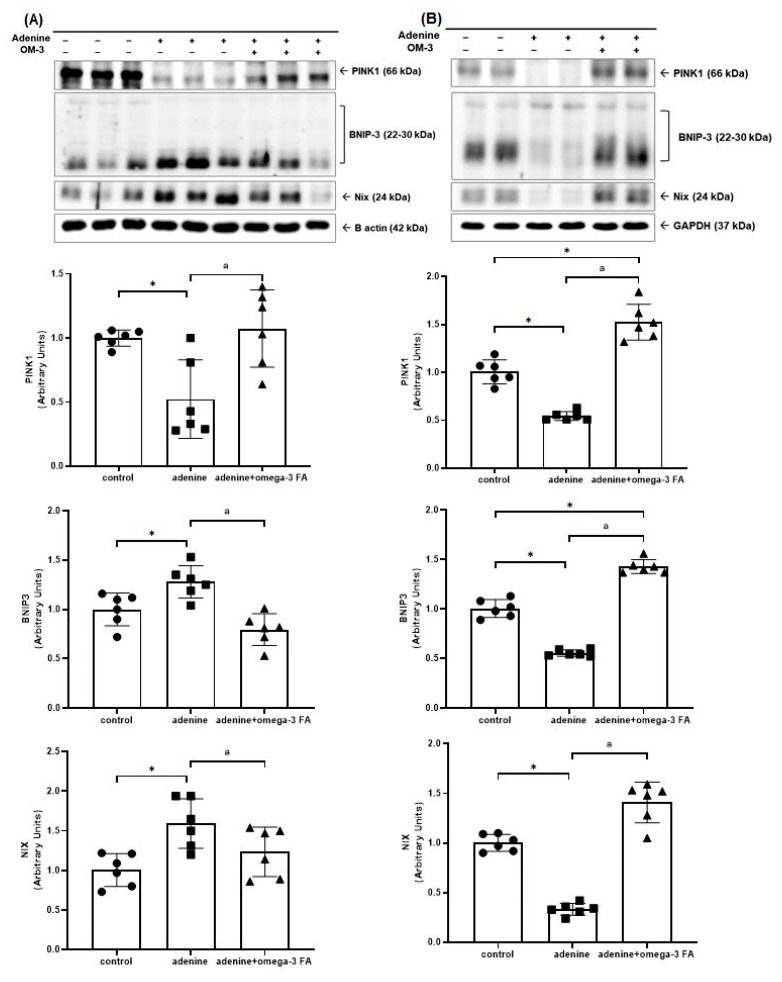
Changes in the expression of factors related to mitochondrial mitophagy including PINK1, BNIP3, and NIX in the kidney (**A**) and heart (**B**). * *p* value < 0.05 (mean values are significantly different from the control group). ^a^
*p* value < 0.05 (mean values are significantly different from the adenine group).

**Figure 4 biomedicines-12-02107-f004:**
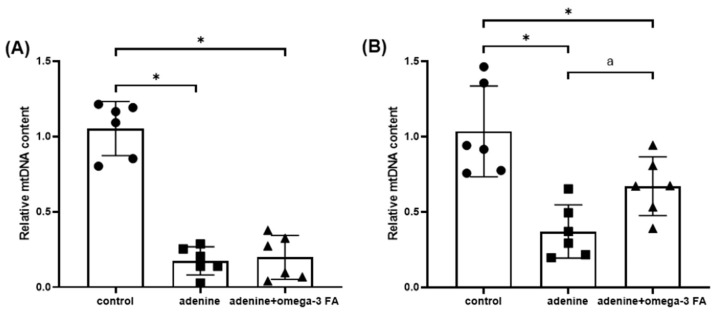
Relative mitochondrial DNA (mtDNA) content in the kidney (**A**) and heart (**B**). * *p* value < 0.05 (mean values are significantly different from the control group). ^a^
*p* value < 0.05 (mean values are significantly different from the adenine group).

**Table 1 biomedicines-12-02107-t001:** Characteristics of rats in experimental groups.

	Normal Control (*n* = 6)	Adenine Control (*n* = 6)	Adenine with Omega-3 FA (*n* = 6)	*p* Value
Blood Urea Nitrogen (mg/dL)	10.2 ± 4.8	217.2 ± 161.5 *	127.7 ± 124.4	0.011
Creatinine (mg/dL)	0.5 ± 0.1	6.2 ± 2.0 *	5.1 ± 2.0 *	<0.001
Calcium (mg/dL)	11.4 ± 0.3	9.1 ± 1.2 *	8.8 ± 1.7 *	0.002
Phosphorus (mg/dL)	9.4 ± 1.3	34.6 ± 13.1 *	28.5 ± 9.8 *	<0.001
Kidney weight (mg)-to-body weight (g) ratio	0.71 ± 0.05	3.42 ± 0.46 *	3.54 ± 0.61 *	<0.001
Heart weight (mg)-to-body weight (g) ratio	0.33 ± 0.03	0.48 ± 0.05 *	0.43 ± 0.04 *	0.005

Data are expressed as means ± SD. * *p* value < 0.05 (mean values are significantly different from the normal control group).

## Data Availability

The data that support the findings of this study are available from the corresponding author.

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
