# Peer review of "Omega-3 Fatty Acids Modify Drp1 Expression and Activate the PINK1-Dependent Mitophagy Pathway in the Kidney and Heart of Adenine-Induced Uremic Rats"

_biomedicines, 2024, doi:10.3390/biomedicines12092107_

Round 1
Reviewer 1 Report
Comments and Suggestions for Authors
Introduction
Authors should add information about prevalence of CKD and heart failure in studied areas
Also, add information about role of Omega 3 in treatment of kidney disease or heart failure
Materials
The condition of PCR should be added and content of mastermix
Results
Results are well presented and described
Discussion
The data well discussed with previous studies and presented
Comments on the Quality of English LanguageMinor editing of English language required.
Author Response
Reviewer (#1)' Comments to Author:
Introduction
Authors should add information about prevalence of CKD and heart failure in studied areas
Also, add information about role of Omega 3 in treatment of kidney disease or heart failure
Response: The authors appreciate the reviewer’s comments. Authors added it in the revised manuscript.
“In accordance with the 2022 Korea Health Statistics, the prevalence of CKD aged over 65 years is close to 18% compared to 7% in total adult patients with CKD aged over 18 years [6]. Moreover, the prevalence of HF with CKD is 15.8% of the total Korean population in 2020, compared to 2.58% in patients without CKD [7].”
“Omega-3 fatty acids (FAs) have anti-inflammatory and renoprotective effects against kidney diseases with mitigating oxidative stress and inflammation. Moreover, high level of omega-3 FA was associated with a slower annual decline in renal function. [18-20]. They reduce cardiovascular risk through various mechanisms, including anti-inflammatory action, lowering triglyceride-rich lipoproteins, and antithrombotic effects, with regard to suppress the expression of pro-inflammatory cytokines and infiltration of inflammatory cells [21].”
Materials
The condition of PCR should be added and content of mastermix
Response: The authors appreciate the reviewer’s comments. Authors added it in the revised manuscript.
“The Master Mix contains SYBR Green dye, Dual-Lock Taq DNA Polymerase, dNTPs with dUTP/dTTP blend, heat-labile UDG, ROX passive reference dye, and optimized buffer components. The PCR condition for standard cycling was set according to SYBR Green Master Mix protocol and guidance.”
Results
Results are well presented and described
Response: Thank you very much for your opinion.
Discussion
The data well discussed with previous studies and presented
Response: Thank you very much for your opinion.

Reviewer 2 Report
Comments and Suggestions for Authors
The authors investigated omega-3 fatty acids (FA) to regulate mitochondrial biogenesis, dynamics and mitophagy in kidney and heart of adenine-induced uremic rats, with a large set of assays. The proposed manuscript is innovative, well-written, in stable English, and the proliferator-activated receptor gamma coactivator-1 alpha (PGC-1α), dynamin protein 1 (Drp1), phosphatase and tensin homolog-induced putative kinase 1 (PINK1) used are indicative for the accuracy of the conducted experiment. The research conducted is entirely cliché-oriented, with the authors specifying that Omega-3 fatty acids may be beneficial for mitochondrial homeostasis by activating mitochondrial biogenesis and PINK1-dependent mitophagy in kidney and heart.
Brief notes:
The abstract is well laid out, but the introduction of a graphical abstract or implementation scheme would increase the readability and value of the article.
1. line 62...to add information about the role of omega-3 fatty acids (FA)
2. table 1 is presented very comprehensively.
3. compared to that of the control group. In the adenine with omega-3 FA group, 137....""compared to controls"". The presence of similar to be shortened throughout the text.
4. Figure 1, 2, 3 are illegible. To be re-introduced. To be presented after the relevant results.
5. line 203-207 to be revised
6. nowhere are the oxidative changes at the mitochondrial level related to all the investigated indicators affected.
7.omega-3 FA supplementation / PGC-1α to schematically present a probable mechanism of action.
8. The conclusion to be revised and enriched.
9. To present limitations and future projects.
10. Only 17 references are from the last 5 years. Where possible use new sources.
Comments on the Quality of English Language-
Author Response
The authors investigated omega-3 fatty acids (FA) to regulate mitochondrial biogenesis, dynamics and mitophagy in kidney and heart of adenine-induced uremic rats, with a large set of assays. The proposed manuscript is innovative, well-written, in stable English, and the proliferator-activated receptor gamma coactivator-1 alpha (PGC-1α), dynamin protein 1 (Drp1), phosphatase and tensin homolog-induced putative kinase 1 (PINK1) used are indicative for the accuracy of the conducted experiment. The research conducted is entirely cliché-oriented, with the authors specifying that Omega-3 fatty acids may be beneficial for mitochondrial homeostasis by activating mitochondrial biogenesis and PINK1-dependent mitophagy in kidney and heart.
Brief notes:
The abstract is well laid out, but the introduction of a graphical abstract or implementation scheme would increase the readability and value of the article.
- line 62...to add information about the role of omega-3 fatty acids (FA)
Response: The authors appreciate the reviewer’s comments. Authors added it in the revised manuscript.
“Omega-3 fatty acids (FAs) have anti-inflammatory and renoprotective effects against kidney diseases with mitigating oxidative stress and inflammation. Moreover, high level of omega-3 FA was associated with a slower annual decline in renal function. They reduce cardiovascular risk through various mechanisms, including anti-inflammatory action, lowering triglyceride-rich lipoproteins, and antithrombotic effects, with regard to suppress the expression of pro-inflammatory cytokines and infiltration of inflammatory cells.”
- table 1 is presented very comprehensively.
Response: Thank you very much for your opinion.
- compared to that of the control group. In the adenine with omega-3 FA group, 137....""compared to controls"". The presence of similar to be shortened throughout the text.
Response: Thank you for your comment. As suggested by the reviewer, authors have changed it in the revised manuscript.
- Figure 1, 2, 3 are illegible. To be re-introduced. To be presented after the relevant results.
Response: Thank you for your comment. As suggested by the reviewer, authors have changed and added it in the revised manuscript.
- line 203-207 to be revised
Response: Thank you for pointing this out. we revised the sentence as follows:
“Likewise, previous studies using Drp1 knockout mice have shown that Drp1 downregulation induced mitochondrial dysfunction, apoptosis, left ventricular dysfunction, and heart failure [15,23]. In this study, Drp1 in the heart of adenine-induced CKD mice was down-regulated, and mitigated after omega-3 FA supplementation”
- nowhere are the oxidative changes at the mitochondrial level related to all the investigated indicators affected.
Response: Thank you for pointing this out. Unfortunately, we have not evaluated factors about the oxidative changes at the mitochondrial level. The authors described this limitation in the discussion section.
“There are several limitations to this study. First, we did not evaluate the reactive oxygen species (ROS) level and inflammatory markers related to the oxidative change at the mitochondrial levels in this study.”
7.omega-3 FA supplementation / PGC-1α to schematically present a probable mechanism of action.
Response: Thank you for your comment. The authors described a probable mechanism of action related to omega-3 FA supplementation / PGC-1α in the discussion section of the revised manuscript.
“Our previous study showed that omega-3 FA may improve mitochondrial biogenesis by upregulating Nrf1 and Nrf2. The probable mechanism was related to increased PGC-1α expression and deacetylation of PGC-1α, which was triggered by increased SIRT1/3 production[22].”
- The conclusion to be revised and enriched.
Response: Thank you for your comment. As suggested by the reviewer, the authors have revised and enriched it in the revised manuscript.
“In conclusion, our results suggest that Drp1, which is involved in mitochondrial fission, may be an important target for management in the uremic kidney and heart. Omega-3 FA may be beneficial for mitochondrial homeostasis by activating mitochondrial biogenesis and PINK1-dependent mitophagy in the kidney and heart of adenine-induced uremic rats. Further studies including mitochondrial function, ROS, and glycolytic regulation are necessary to demonstrate that omega-3 FA may be beneficial for preventing the progression of renal function decline and heart failure.”
- To present limitations and future projects.
Response: Thank you very much for your comments. The authors added limitations and future projects in the revised manuscript.
“There are several limitations to this study. First, we did not evaluate the reactive oxygen species (ROS) level and inflammatory markers related to the oxidative change at the mitochondrial levels in this study. Second, we did not perform the pathologic examination to assess the usefulness of omega-3 FA in this CKD model. Third, we have not investigated the glycolytic regulation and dysregulation related to mitochondrial dysfunction and mitochondrial homeostasis.”
“Further studies including mitochondrial function, ROS, and glycolytic regulation are necessary to demonstrate that omega-3 FA may be beneficial for preventing the progression of renal function decline and heart failure.”
- Only 17 references are from the last 5 years. Where possible use new sources.
Response: Thank you very much for your comments. The authors added new recent references in the revised manuscript.

Reviewer 3 Report
Comments and Suggestions for Authors
This is an interesting and competent study. In proper context, it should impart new understanding, and seems to steer clear of over-interpretation.
Conversely, however, the manuscript is very narrow, since it somehow manages to avoid discussing the PINK-1 pathway in the context of glycolytic regulation (or dysregulation) which is substantially influenced by adenine levels, with corresponding roles in mitochondrial dysfunction, including effects on mitochondrial homeostasis, plus induction of the Warburg effect which, in turn, is considered to play a significant role in chronic kidney disease. (Apologies for the run-on sentence.)
It seems to me that the manuscript would be better served by dedicating some text to placing these findings within the context of key pathological aspects of CKD, so that nephrologists can better grasp the implications of this study.
There is a huge volume of additional recent literature, not cited herein, that could be commented upon to help to better establish context. It's hard to know where to start, but maybe by suggesting that the authors consider a couple of very recent Nature papers:
The mitophagy pathway and its implications in human diseases.
https://www.nature.com/articles/s41392-023-01503-7
Mitochondrial metabolic reprogramming in diabetic kidney disease.
https://www.nature.com/articles/s41419-024-06833-0
Author Response
Reviewer (#3)' Comments to Author:
This is an interesting and competent study. In proper context, it should impart new understanding, and seems to steer clear of over-interpretation.
Conversely, however, the manuscript is very narrow, since it somehow manages to avoid discussing the PINK-1 pathway in the context of glycolytic regulation (or dysregulation) which is substantially influenced by adenine levels, with corresponding roles in mitochondrial dysfunction, including effects on mitochondrial homeostasis, plus induction of the Warburg effect which, in turn, is considered to play a significant role in chronic kidney disease. (Apologies for the run-on sentence.)
Response: Thank you for pointing this out. We have not investigated about the glycolytic regulation and dysregulation. The authors described this limitation and perspectives in the discussion section.
“Third, we have not investigated the glycolytic regulation and dysregulation related to mitochondrial dysfunction and mitochondrial homeostasis.”
It seems to me that the manuscript would be better served by dedicating some text to placing these findings within the context of key pathological aspects of CKD, so that nephrologists can better grasp the implications of this study.
Response: Thank you for pointing this out. Unfortunately, we have not evaluated key pathological aspects of CKD. The authors described this limitation in the discussion section.
“Second, we did not perform the pathologic examination to assess the usefulness of omega-3 FA in this CKD model.”
There is a huge volume of additional recent literature, not cited herein, that could be commented upon to help to better establish context. It's hard to know where to start, but maybe by suggesting that the authors consider a couple of very recent Nature papers:
The mitophagy pathway and its implications in human diseases.
https://www.nature.com/articles/s41392-023-01503-7
Mitochondrial metabolic reprogramming in diabetic kidney disease.
https://www.nature.com/articles/s41419-024-06833-0
Response: Thank you very much for your comments. Authors added important two references in the revised manuscript.

Round 2
Reviewer 3 Report
Comments and Suggestions for Authors
I remain of the impression that there are limits to the impact of a study about mitophagy that pays so little attention to the dysfunctional mitochondria that are targeted in mitophagy. Nonetheless, the study may contribute to the overall knowledge in the field, and does not suffer from other obvious limitations, so I will recommend it for publication.